# Measuring causal influence with back-to-back regression: the linear case

## Abstract

Identifying causes from observations can be particularly challenging when i) potential factors are difficult to manipulate individually and ii) observations are complex and multi-dimensional. To address this issue, we introduce "Back-to-Back" regression (B2B), a method designed to efficiently measure, from a set of co-varying factors, the causal influences that most plausibly account for multidimensional observations. After proving the consistency of B2B and its links to other linear approaches, we show that our method outperforms least-squares regression and cross-decomposition techniques (e.g. canonical correlation analysis and partial least squares) on causal identification. Finally, we apply B2B to neuroimaging recordings of 102 subjects reading word sequences. The results show that the early and late brain representations, caused by low- and high-level word features respectively, are more reliably detected with B2B than with other standard techniques.

## 1 Introduction

Natural sciences are tasked to find, from a set of hypothetical factors, the minimal subset that suffices to reliably predict novel observations. This endeavor is impeded by two major challenges.

First, causal and non-causal factors may be numerous and collinear. This issue becomes increasingly pronounced as the number of potential factors increases. In neuroscience, for example, identifying whether the frequency of a word presented on a subject's retina modulates brain activity can be surprisingly difficult. Indeed, the frequency of words in natural language covaries with other factors such as their length (short words are more frequent than long words), their categories (determinants are more frequent than adverbs) and so forth (Kutas & Federmeier, 2011; Pegado et al., 2014). Instead of selecting a set of words that control for all of these factors simultaneously, it is thus common to use forward modeling, i.e. to train a model to predict observations (e.g. brain activity) from a minimal combination of competing factors (e.g. word length, word frequency, e.g. (Huth et al., 2016)), and investigate, in the model, the estimated contribution of each factor (Friston et al., 1994).

The second challenge to measuring causal influence is that observations can be large and complex. The relationship between causes and effects is thus often considered in a backward manner, by training models to maximally predict causes from multidimensional observations. For example, brain activity is often recorded with hundreds or thousands of sensors simultaneously. As multiple sensors may be affected by common noise sources, it is common to use backward regression, by, for example, fitting a support vector machine across multiple sensors to decode the category of a stimulus (Cichy et al., 2014; Kriegeskorte et al., 2008; Norman et al., 2006).

Both forward and backward modeling have competing benefits and drawbacks. Specifically, forward modeling disentangles the independent contribution of collinear factors, but does not combine multidimensional observations. By contrast, backward modeling combines multiple observations, but does not disentangle collinear factors (Weichwald et al., 2015; Hebart & Baker, 2018; King et al., 2018). To combine the benefits of forward and backward modeling, several authors have proposed to use cross-decomposition techniques such as Partial Least Squares (PLS) and Canonical Correlation Analysis (CCA) (de Cheveigne et al., 2019). CCA and PLS aim to find, from two sets of data $X$ and $Y$, the components $H$ and $G$ were $XH$ and $YG$ are maximially correlated or maximally covarying respectively. Because CCA and PLS are based on a generalized eigen decomposition, their resulting coefficients are mixing the features of $X$ and $Y$ in a way that makes them notoriously difficult to interpret (Lebart et al., 1995).

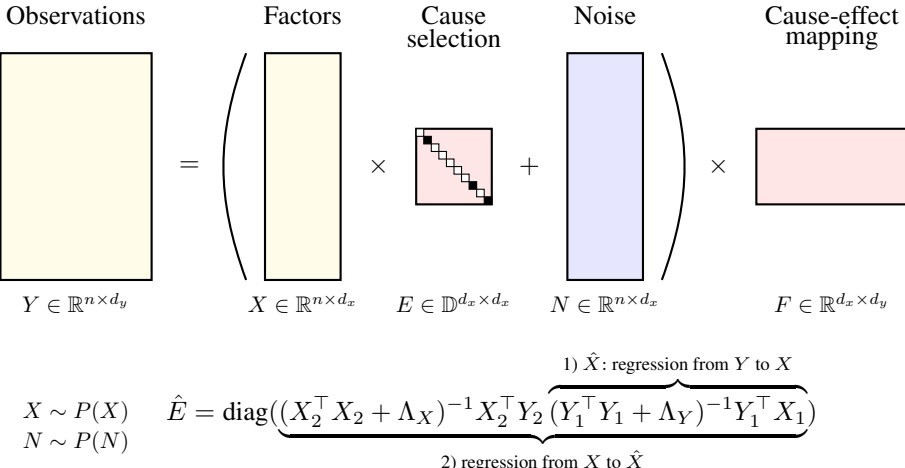

Figure 1: Back-to-back regression identifies the subset of factors $E_{ii} = 1$ in $X$ that influence some observations $Y$ by 1) regressing from $Y$ to $X$ to obtain $\hat{X}$, and 2) returning the diagonal of the regression coefficients from $X$ to $\hat{X}$.

Here, we introduce the 'back-to-back regression' (B2B), which not only combines the benefits of forward and backward modeling (Section 2), but also provide robust, interpretable and unidimensional causal coefficients for each tested factor.

The present paper focuses on the restricted issue of disentangling the causal influence of linearly correlated predictors ($X$) onto noisy multivariate observations ($Y$). The present approach thus differs from other causal discovery algorithms based on temporal-delays and/or nonlinear interactions in systems where the directionality of causation (from X to Y or vice versa) is unknown (e.g. (Peters et al., 2017; Granger, 1969; Janzing et al., 2013; Schölkopf et al., 2016).

After detailing B2B method and proving its convergence (Section 2.2), we show with synthetic data that it outperforms state-of-the-art forward, backward and cross-decomposition techniques in identifying causal influence (Section 3.1). Finally, we apply B2B to a large neuroimaging dataset and reveal that distinct but collinear word features lead to distinguishable brain representations (Section 3.5).

## 2 BACK-TO-BACK REGRESSION

We consider the measurement of multivariate signal $Y \in \mathbb{R}^{n \times d_y}$, generated from a set of putative causes $X \in \mathbb{R}^{n \times d_x}$, via some unknown linear apparatus $F \in \mathbb{R}^{d_x \times d_y}$. Not all the variables in $X$ exert a causal influence on $Y$. By considering a square binary diagonal matrix of *causal influences* $E \in \mathbb{D}^{d_x \times d_x}$, we denote by $XE$ the causal factors of $Y$. In summary, the problem can be formalized as:

$$y_i = (x_i E + n_i) F \tag{1}$$

where $i$ is a given sample, and $n_i$ is a sample-specific noise drawn from a centered distribution. While the triplet of variables $X$ and $N$ are independent, we allow each of them to have any form of covariance. In practice, we observe $n$ samples $(X, Y)$ from the model. This problem space, along with the sizes of all variables involved, is illustrated in Figure 1. Given the model in Equation (1), **the goal** of Back-to-Back Regression (B2B) is to estimate the matrix of causal influences $E$.

### 2.1 ALGORITHM

Back-to-Back Regression (B2B) consists of two steps. First, we estimate the linear regression coefficients $\hat{G}$ from $Y$ to $X$, and construct the predictions $\hat{X} = Y\hat{G}$. This backward regression recovers the correlations between $Y$ and each factor of $X$. Second, we estimate the linear regression

coefficients $\hat{H}$ from $X$ to $\hat{X}$. The diagonal of the regression coefficients $\hat{H}$, denoted by $\hat{E} = \mathrm{diag}(\hat{H})$, is the desired estimate of the causal influence matrix $E$, as detailed in the A.1.

If using l2-regularized least-squares (Hoerl, 1959; Rifkin & Lippert, 2007), B2B has a closed form solution:

$$\hat{G} = (Y^\top Y + \Lambda_Y)^{-1} Y^\top X, \tag{2}$$

$$\hat{H} = (X^\top X + \Lambda_X)^{-1} X^\top Y \hat{G}, \tag{3}$$

where $\Lambda_X$ and $\Lambda_Y$ are two diagonal matrices of regularization parameters, useful to invert the covariance matrices of $X$ and $Y$ if these are ill-conditioned.

Performing two regressions over the same data sample can result in overfitting, as spurious correlations in the data absorbed by the first regression will be leveraged by the second one. To avoid this issue, we split our sample $(X, Y)$ into two splits $(X_1, Y_1)$ and $(X_2, Y_2)$. Then, the first regression is performed using $(X_1, Y_1)$, and the second regression is performed using $(X_2, Y_2)$. To compensate for the reduction in sample size caused by the split, B2B is repeated over many random splits, and the final estimate $\hat{E}$ of the causal influence matrix is the average over the estimates associated to each split (Breiman, 1996). To accelerate this ensembling procedure, we implemented an efficient leave-one-out cross-validation scheme as detailed in (Rifkin & Lippert, 2007), as follows:

$$\hat{Y} = (\Sigma_X G Y - \mathrm{diag}(\Sigma_X G)Y) \,/\, \mathrm{diag}(I - \Sigma_X G) \qquad \text{(element-wise division)} \tag{4}$$

where $\Sigma_X$ is the $X$ kernel matrix and where $G$ is computed with an eigen decomposition of $X$:

$$\Sigma_X = QVQ^T$$
$$G = Q(V + \lambda I)^{-1} Q^T \tag{5}$$

where $Q$, $V$ and $\lambda$ are the eigen vectors, eigen values and regularization, respectively.

We summarize the B2B procedure in Algorithm 1. The rest of this section provides a theoretical guarantee on the correctness of B2B.

---

**Algorithm 1:** Back-to-back regression.

    **Input:** input data $X \in \mathbb{R}^{n \times d_x}$, output data $Y \in \mathbb{R}^{n \times d_y}$, number of repetitions $m \in \mathbb{N}$.
    **Output:** estimate of causal influences $\hat{E} \in \mathbb{D}^{d_x \times d_x}$.

1   $\hat{E} \leftarrow 0$;
2   **for** $i = 1, \dots, m$ **do**
3     $(X, Y) \leftarrow \text{ShuffleRows}((X, Y))$;
4     $(X_1, Y_1), (X_2, Y_2) \leftarrow \text{SplitRowsInHalf}((X, Y))$;
5     $\hat{G} = \text{LinearRegression}(Y_1, X_1)$ ;            $\triangleright$   $\hat{G} = (Y_1^\top Y_1 + \Lambda_Y)^{-1} Y_1^\top X_1$
6     $\hat{H} = \text{LinearRegression}(X_2, Y_2 \hat{G})$ ;       $\triangleright$   $\hat{H} = (X_2^\top X_2 + \Lambda_X)^{-1} X_2^\top Y_2 \hat{G}$
7     $\hat{E} \leftarrow \hat{E} + \mathrm{diag}(\hat{H})$;
8   **end**
9   $\hat{E} \leftarrow \hat{E}/m$;
10   $\hat{W} \leftarrow \text{LinearRegression}(X\hat{E}, Y)$;
11   **return** $\hat{E}, \hat{W}$

---

## 2.2 THEORETICAL GUARANTEES

**Theorem 1** (B2B consistency - general case). *Consider the B2B model from Equation $Y = (XE + N)F$, $N$ centred and full rank noise. Let $Img(M)$ refers to the image of the matrix $M$. If $F$ and $X$ are full-rank on the $Img(E)$, then, the solution of B2B, $\hat{H}$, will minimize $\min_H \|X - XH\|^2 + \|NH\|^2$ and satisfy $E\hat{H} = \hat{H}$*

*Proof.* See Appendix A.1.                    □

Since $E\hat{H} = \hat{H}$, we have

$$\hat{H} = \arg\min_H \|X - XEH\|^2 + \|NEH\|^2 = (EX^\top XE + EN^\top NE)^\dagger EXX^\top. \tag{6}$$

Assuming, without loss of generality, that the active features in $E$ are the $k \in \mathbb{Z} : k \in [0, d_x]$ first features, and rewriting $X = (X_1, X_2)$ and $N = (N_1, N_2)$ ($X_1$ and $N_1$ containing the $k$ first features), we have:

$$X^\top X = \begin{pmatrix} \Sigma_{X_1 X_1} & \Sigma_{X_1 X_2} \\ \Sigma_{X_1 X_2} & \Sigma_{X_2 X_2} \end{pmatrix}, \qquad N^\top N = \begin{pmatrix} \Sigma_{N_1 N_1} & \Sigma_{N_1 N_2} \\ \Sigma_{N_1 N_2} & \Sigma_{N_2 N_2} \end{pmatrix}, \tag{7}$$

where $\Sigma_{AB}$ is the covariance of $A$ and $B$, and:

$$\hat{H} = \begin{pmatrix} (\Sigma_{X_1 X_1} + \Sigma_{N_1 N_1})^{-1} \Sigma_{X_1 X_1} & (\Sigma_{X_1 X_1} + \Sigma_{N_1 N_1})^{-1} \Sigma_{X_1 X_2} \\ 0 & 0 \end{pmatrix} \tag{8}$$

$$\mathrm{diag}_k(\hat{H}) = \mathrm{diag}((\Sigma_{X_1 X_1} + \Sigma_{N_1 N_1})^{-1} \Sigma_{X_1 X_1}) = \mathrm{diag}((I + \Sigma_{X_1 X_1}^{-1} \Sigma_{N_1 N_1})^{-1}) \tag{9}$$

In the absence of noise, we have $\Sigma_{N_1 N_1} = 0$, and so $\mathrm{diag}_k(\hat{H}) = I$, and

$$\mathrm{diag}(\hat{H}) = \mathrm{diag}(E)$$

Therefore, we recover $E$ from $\hat{H}$.

In the presence of noise, the causal factors of $E$ correspond to the positive elements of $\mathrm{diag}(\hat{H})$. The methods to recover them are presented in the Appendix A.4.

## 3 EXPERIMENTS

We perform two sets of experiments to evaluate B2B: one on controlled synthetic data, and a second one on a real, large-scale magneto-encephalography (MEG) dataset. We use scikit-learn's PLS and RidgeCV (Pedregosa et al., 2011) as well as pyrcca's regularized canonical component analysis (RegCCA, (Bilenko & Gallant, 2016)) objects to compare B2B against the standard baselines.

### 3.1 SYNTHETIC DATA

We evaluate the performance of B2B throughout a series of experiments on controlled synthetic data. The purpose of these experiments is to evaluate the ability of B2B in terms of prediction of independent and identically distributed data, as well as a method to recover causal factors.

The data generating process for each experiment constructs $n = 1000$ training examples according to the model $Y = (\mathrm{h}XE + N)F$, where h is a scalar that modulates the signal-to-noise ratio. Here, $F \in \mathbb{R}^{d_x \times d_y}$ contains entries drawn from $\mathcal{N}(0, \sigma^2)$ where $\sigma^2$ is inversely proportional to $d_x$, $X \in \mathbb{R}^{n \times d_x}$ contains rows drawn from $\mathcal{N}(0, \Sigma_X)$, $N \in \mathbb{R}^{n \times d_x}$ contains rows drawn from $\mathcal{N}(0, \Sigma_N)$, $E \in \mathbb{R}^{d_x \times d_x}$ is a binary diagonal matrix containing $n_c$ ones, $\Sigma_X = AA^\top$ where $A \in \mathbb{R}^{d_x \times d_x}$ contains entries drawn from $\mathcal{N}(0, \sigma^2)$, $\Sigma_N = BB^\top$ where $B \in \mathbb{R}^{d_x \times d_x}$ contains entries drawn from $\mathcal{N}(0, \sigma^2)$, and the factor $\mathrm{h} \in \mathcal{R}_+$.

To simulate a wide range of experimental conditions, we sample 10 values in log-space for $d_x, d_y \in [10, 100]$, $n_c \in [3, 63]$, $\mathrm{h} \in [0.001, 10]$. We discard the cases where $n_c > d_x$, limit $d_x, d_y$ to 100 to keep the running time under 2 hours for each condition, and average over 5 random seeds.

We compare the performance of B2B against four competing methods, all implemented in scikit-learn (Pedregosa et al., 2011) and pyrcca (Bilenko & Gallant, 2016):

### 3.2 BASELINE MODELS

Forward regression consists of an $l2$-regularized "ridge" regression from the putative causes $X$ to the observations $Y$:

$$H_{fwd} = (X^T X + \lambda I)^{-1} X^T Y \tag{10}$$

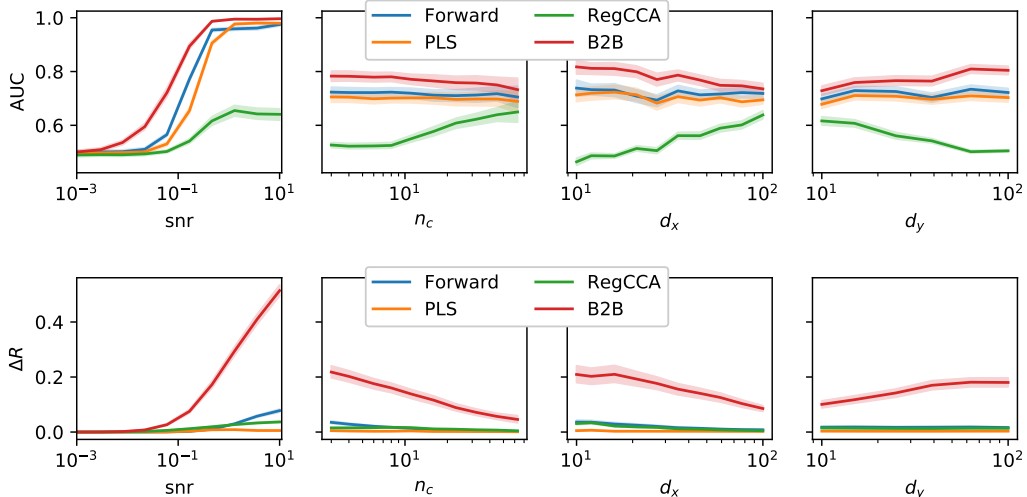

Figure 2: Synthetic experiments. Average AUC (top) and Feature Importance $\Delta R$ (bottom) when varying experimental conditions individually. Higher is better. B2B compares favorably in all cases.

Backward regression consists of an $l2$-regularized "ridge" regression from $Y$ to $X$:

$$G_{bwd} = (Y^T Y + \lambda I)^{-1} Y^T X \tag{11}$$

CCA finds $G_{cca} \in \mathbb{R}^{d_z, d_y}$ and $H_{cca} \in \mathbb{R}^{d_z, d_x}$ s.t. $X$ and $Y$ are maximally correlated in a latent $Z$ space:

$$G_{cca}, H_{cca} = \underset{G, H}{\operatorname{argmax}} \, corr(XH^T, YG^T) \tag{12}$$

PLS finds $G_{pls} \in \mathbb{R}^{d_z, d_y}$ and $H_{pls} \in \mathbb{R}^{d_z, d_x}$ s.t. $X$ and $Y$ are maximally covarying in a latent $Z$ space:

$$G_{pls}, H_{pls} = \underset{G, H}{\operatorname{argmax}} \, cov(XH^T, YG^T) \tag{13}$$

We employ 5-fold cross-validation to select the optimal number of components for CCA and PLS. Regressions were $\ell 2$-regularized with a $\lambda$ regularization parameters fitted with the efficient leave-one-out procedure implemented in scikit-learn RidgeCV (Pedregosa et al., 2011).

### 3.3 EVALUATING CAUSAL DISCOVERY FROM MODELS' COEFFICIENTS

B2B leads to unbiased (i.e. zeros-centered) scalar coefficients for non-causal features. In contrast, the Forward, Backward, CCA and PLS models lead to a loading vector $H_i$ per feature $i$ (or one vector $G^i$ for the backward model). To transform such vector into an estimated causal contribution $\hat{E}$, we take the sum of square coefficients: $\hat{E}_i = \sum_j H_i^{j\,2}$

To estimate whether models accurately identify causal factors, we compute the area-under-the-curve (AUC) across factors $AUC(E, \hat{E})$. The AUC allows evaluating the capacity of models at detecting the causal importance of factors when ground truth labels are available, as is the case in this setup.

We report AUC results in Figures 2 (top) and 5 (left, in Appendix), and compare favorably to all baselines.

### 3.4 EVALUATING CAUSAL DISCOVERY WITH HELD-OUT PREDICTION RELIABILITY

In most cases, $E$ is not known and AUC can thus not be estimated. To address this issue, we assess the ability of each model to reliably predict independent and identically distributed data from $Y$, given all of the $X$ features versus all-but-ones feature $X_{-i}$ (i.e. 'knock-out X'). This procedure

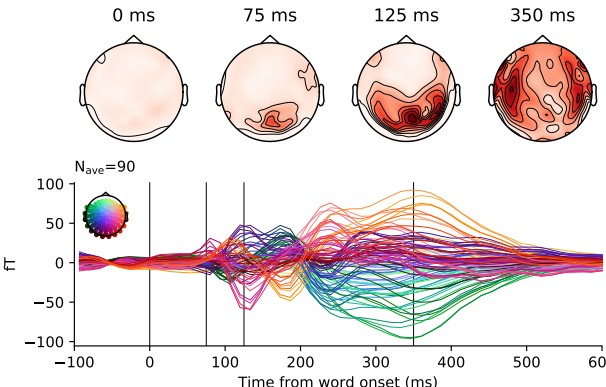

Figure 3: Ninety subjects each read approximately 2,700 words while their brain activity was recorded with MEG. Top. Average brain response to words (word onset at t=0 ms), as viewed from above the head (red= higher gradient of magnetic flux). Bottom. Each line represents a magnetometer, color-coded by its spatial position. Posterior responses, typical of primary visual cortex activity, peak around 100 ms after word onset and are followed by an anterior propagation of activity typical of semantic processing in the associative cortices.

results in two correlation metrics $R_{full}$ and $R_{knockout}$, whose difference $\Delta R_i = R_{full} - R_{knockout}$ indicates how much each $X_i$ improves the prediction of $Y$. In our figures, $\Delta R$ is the average of $\Delta R_i$. A higher score means that for prediction, the model relies on individual features rather than combinations of features.

We show in Appendix A.3 pseudo-code to assess feature importance for our algorithm as well as baselines. For the Backward Model, feature importance cannot be assessed as the $X$ colinearity is never taken into account.

We show in Figures 2 (bottom) and 5 (right, in Appendix) that our method outperforms baselines.

## 3.5 MAGNETOENCEPHALOGRAPHY DATA

Next, we apply our method to brain imaging data from the anonymized multimodal neuroimaging "Mother Of all Unification Studies" (MOUS) dataset (Schoffelen et al., 2019). The dataset contains magneto-encephalography (MEG) recordings of 102 healthy native-Dutch adults who participated in a reading task. Twelve subjects were excluded from the analysis because of corrupted file headers. Subjects were exposed to a rapid serial visual presentation of Dutch words. The word lists consisted of 120 sentences, and scrambled lists of the same words. Each word was presented on the computer screen for 351ms on average (min: 300ms, max: 1400ms). Successive words were separated by a blank screen for 300ms, and successive sentences were separated by an empty screen for a few (3-4) seconds.

### 3.5.1 MEG PREPROCESSING

The raw MEG data was bandpass-filtered between 0.1 and 40Hz using MNE-Python default parameters (Gramfort et al., 2013; 2014). Specifically, we used a zero-phase finite impulse response filter (FIR) with a Hamming window and with transition bands of 0.1Hz and 10Hz for the low and high cut-off frequencies. The raw data was then segmented 100ms before word onset and 1s after word onset ($t = 0$ms corresponds to word onset). Finally, each resulting segment was baseline-corrected between -100ms and 0ms, and decimated by 5 and thus led a sampling frequency of 240Hz. The average responses across words is displayed in Figure 3. For each subject and each time sample relative to word onset, we build an observation matrix $Y \in \mathbb{R}^{n \times d_y}$ of $n \approx 2700$ words by $d_y = 301$ MEG channels (273 magnetometers and 28 compensation channels). Each of the columns of $Y$ is normalized to have zero mean and unit variance.

### 3.5.2 FEATURE DEFINITION

We aim to identify the word features that cause a variation in brain responses. We consider four distinct but colinear features. First, 'Word Length' refers to the total number of letters. Word Length is expected to specifically cause a variation in the early evoked MEG responses (i.e. from 100 ms after stimulus onset) elicited by the retinotopically-tuned visual cortices (e.g. (Pegado et al., 2014).). Second, 'Word Frequency' indexes how frequently each word appears in Dutch and was derived with the the Zipf logarithmic scale of (Van Heuven et al., 2014) provided by the WordFreq

package (Speer et al., 2018). Word Frequency is expected to specifically cause a variation in the late evoked MEG responses (i.e. from 400 ms), because it variably engages semantic processes in the temporal cortices (Kutas & Federmeier, 2011). Third, 'Word Function' indicates whether each word is a content word (i.e. a noun, a verb, an adjective or an adverb) or a function word (i.e. a preposition, a conjunction, a determinant, a pronoun or a numeral), and was derived from Spacy's part of speech tagger (Honnibal & Montani, 2017). To our knowledge, this feature has not been thouroughly investigated with MEG. Its causal contribution to reading processes in the brain thus remains unclear. Finally, to verify that B2B and other methods would not inadequately identify non-causal features, we added a dummy feature, constructed from a noisy combination of Word Length and Word Frequency: $dummy = z(length) + z(frequency) + \mathcal{N}$, where $z$ normalizes features and $\mathcal{N}$ is a random vector sampling Gaussian distribution (all terms thus have a zero-mean and a unit-variance). This procedure yields an $X \in \mathbb{R}^{n \times d_x}$ matrix of $n \approx 2700$ words by $d_x = 4$ features for each subject. Each of the columns of $X$ is normalized to have a mean and a standard deviation of 0 and 1 respectively.

### 3.5.3 MODELS AND STATISTICS

We compare B2B to four standard methods: Forward regression, Backward regression, CCA and PLS, as implemented in scikit-learn (Pedregosa et al., 2011) and (Bilenko & Gallant, 2016), and optimized with nested cross-validation over twenty $l2$ regularization parameters logarithmically spaced between $10^{-4}$ and $10^4$ (for regression and CCA methods) or 1 to 4 canonical components (for PLS).

We used the feature importance described in Algorithm 2 to assess the extent to which each feature $X_i$ specifically improves the prediction of held-out $Y$ data, using a 5-fold cross-validation (with shuffled trials to homogeneize the distributions between the training and testing splits).

Each model was implemented for each subject and each time sample independently. Pairwise comparison between models were performed using a two-sided Wilcoxon test across subjects (n=90) using the average $\Delta R$ across time. Corresponding effect sizes are shown in Figure 4, and p-values are reported below.

### 3.5.4 RESULTS

We compared the ability of Forward regression, Backward regression, CCA, PLS and B2B to estimate the causal contribution of four distinct but collinear features on brain evoked responses to words.

As expected, the Backward model reveals a similar decoding time course for Word Length and Word Frequency, even though these features are known to specifically influence early and late MEG responses respectively (Kutas & Federmeier, 2011). In addition, the same decoding time course was observed for the dummy variable. These results illustrate that backward modelling cannot be used to estimate the causal contribution of collinear features.

We thus focus on the four remaining methods (i.e. Forward Regression, PLS, CCA, and B2B) and estimate their $\Delta R$ (i.e. the improvement of Y prediction induced by the introduction of a given feature into the model, as described in Algorithm 2). Contrary to the Backward Model, none of the models predicted the Dummy Variable to improve the $Y$ prediction: all $\Delta R < 0$ (all $p > .089$).

Figure 4 shows, for each model, the effects obtained across time (left) and subjects (right).

Word Length and Word Frequency improved the prediction performance of all methods: $\Delta R > 0$ for all models (all $p < 0.0001$). As expected, the time course associated with Word Length and Word Frequency rose from $\approx 100$ ms and from $\approx 400$ ms respectively. Furthermore, Word Function improved the prediction performance of all models (all $p < 0.0002$) except for PLS ($p = 0.7989$). Overall, these results confirm that Word Length, Word Frequency and Word Function causally influence specific periods of brain responses to words.

To assess which model would be most sensitive to these causal discoveries, we compared B2B to other models across subjects (Figure 4 right). For Word Length B2B outperforms all models (all $p < 0.00001$) but CCA ($p = 0.0678$). For Word Frequency, B2B outperforms all models (all $p < 0.0006$). For "Word Function", B2B outperforms all models (all $p < 0.0015$). Overall, these results show that B2B reliably outperforms standard methods, especially when the effects are difficult to detect.

## 4 RELATED WORK

Forward and cross-decomposition models have been used to identify the causal contribution of collinear features onto multi-dimensional observations (e.g. (Naselaris et al., 2011)). These approaches typically lead to multiple coefficients for each features (i.e. one per dimension of $Y$ or one per component respectively). Furthermore, these coefficients can be difficult to summarize into a single causal estimate. By contrast, B2B quickly (Fig. 6) leads to a single unbiased scalar values $\hat{E}$ tending towards 1 and 0 for causal and non-causal features respectively.

A variety of other statistical methods applied to neuroimaging data have been proposed to clarify what is being represented in brain responses - i.e. what feature causes specific brain activity. One of the popular linear method is Representational Similarity Analysis (RSA) (Kriegeskorte et al., 2008), and consists in analyzing the similarity of brain responses associated with specific categorical conditions (e.g. distinct images), by (1) fitting one-against-all classifiers on each condition and (2) testing whether these classifiers can discriminate all other conditions. The resulting confusion matrix is then analyzed in an unsupervised manner to reveal which conditions lead to similar brain activity patterns. B2B differs from RSA in that (1) it uses regressions instead of classifications, and can thus generalize to new items and new contexts and (2) it is fully supervised.

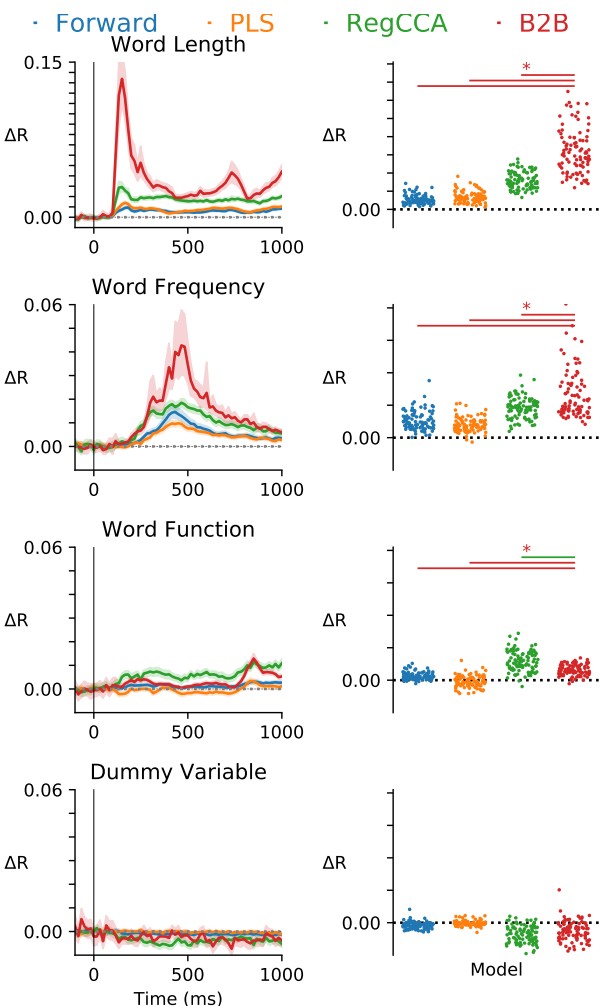

Figure 4: Multiple models (color-coded) are compared on their ability to reliably predict single-trial MEG signals evoked by words. Left. Average improvement of correlation coefficient $\Delta R$ for each of the four features (rows). Error bars indicate standard error of the mean (SEM) across subjects. Right. Average $\Delta R$ across time for each subject (dots). Top horizontal lines indicate when B2B significantly outperforms other methods (red) and vice versa.

Finally, CCA has been used in neuroimaging for a variety of purposes such as denoising and subject alignment (Hotelling, 1936; de Cheveigne et al., 2019). While CCA relates to B2B, these two methods diverge in several ways. First, CCA and B2B have different objectives: CCA aims to find the potentially numerous and poorly interpretable components where X and Y are maximally correlated, whereas B2B aims to recover the causal factors from X to Y. Second, B2B is not symmetric between $X$ and $Y$: it aims to identify specific causal features by first optimizing over the decoders $G$ and then over $H$. By contrast, CCA is symmetric between $X$ and $Y$, and aims to find $G$ and $H$ such that they project $X$ and $Y$ on maximally correlated dimensions. Third, CCA is based an eigen decomposition of $XH$ and $YG$ - the corresponding canonical components are thus mixing the $X$ features in way that limit interpretability and potentially dilute the impact of each feature onto multiple components. In contrast B2B assesses each feature $X^j$ on a single $Y$ component specifically selected to maximize signal-to-noise ratio of that feature $j$. Fourth, and unlike B2B, CCA does not separately optimize two distinct regularization parameters for $G$ and $H$. Finally, CCA does not use different data splits to estimate $G$ and $H$. Together, these differences may explain why B2B reliably outperform CCA on estimating causal influences (Figs. 2 and 5).

## 5  CONCLUSION

In this work, we proposed Back-to-Back (B2B) regression, a linear method to measure the causal influence of a potential set of variables generating multidimensional observations. B2B performs two successive multidimensional regressions: one from the output domain, and another one from the input domain. We provided a theoretical guarantee about the consistency of B2B, and compared it to several baselines in controlled synthetic experiments. We also applied B2B to a recent brain imaging dataset, analyzing the timing of brain responses and their connection to word features. We obtained results consistent with prior work in neuroscience literature, confirming the efficacy of B2B for real data analysis.

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

# A  APPENDIX

## A.1  PROOF OF CONSISTENCY THEOREM

Proof of the theorem in 2.2:

**Theorem 2** (B2B consistency - general case)**.** *Consider the B2B model from equation 1*

$$Y = (XE + N)F$$

*with $N$ centred and full rank noise.*

*If $F$ and $X$ are full-rank on $Img(E)$, then, the solution of B2B, $\hat{H}$ minimizes*

$$\min_H \|X - XH\|^2 + \|NH\|^2$$

*and satisfies*

$$E\hat{H} = \hat{H}$$

*Proof.* Let $\hat{G}$ and $\hat{H}$ be the solutions of the first and second regressions of B2B.

Since $\hat{G}$ is the least square estimator of $X$ from $Y$

$$\hat{G} = \arg\min_G \mathbb{E}[\|YG - X\|^2]$$

Replacing $Y$ by its model definition $Y = (XE + N)F$, we have

$$\hat{G} = \arg\min_G \mathbb{E}[\|X - (XE + N)FG\|^2] = \arg\min_G \mathbb{E}[\|X - XEFG + NFG\|^2]$$

Since $N$ is centered and independent of $X$, we have

$$\hat{G} = \arg\min_G \|X - XEFG\|^2 + \|NFG\|^2 \tag{14}$$

Samely, for $\hat{H}$, we have

$$\hat{H} = \arg\min_H \mathbb{E}[\|XH - Y\hat{G}\|^2] = \arg\min_H \mathbb{E}[\|XH - (XE + N)F\hat{G}\|^2]$$

$$= \arg\min_H \mathbb{E}[\|X(H - EF\hat{G})\|^2] + \mathbb{E}[\|NF\hat{G}\|^2]$$

$$= \arg\min_H \mathbb{E}[\|X(H - EF\hat{G})\|^2]$$

a positive quantity which reaches a minimum (zero) for

$$\hat{H} = EF\hat{G} \tag{15}$$

Let us now prove that $EF\hat{G} = F\hat{G}$.

Let $F^\dagger$ be the pseudo inverse of $F$, and $Z = F^\dagger EF\hat{G}$, we have $FZ = FF^\dagger EF\hat{G}$

Since $F$ is full rank on $Img(E)$, we have $FF^\dagger E = E$, and $FZ = EF\hat{G}$

As $E$ is a binary diagonal matrix, it is an orthogonal projection and therefore a contraction, thus

$$\|NEF\hat{G}\|^2 \le \|NF\hat{G}\|^2$$

and

$$\|X - XEFZ\|^2 + \|NFZ\|^2 = \|X - XEF\hat{G}\|^2 + \|NEF\hat{G}\|^2 \le \|X - XEF\hat{G}\|^2 + \|NF\hat{G}\|^2$$

But since $\hat{G} = \arg\min_G \|X - XEFG\|^2 + \|NFG\|^2$, we also have

$$\left\|X - XEF\hat{G}\right\|^2 + \left\|NF\hat{G}\right\|^2 \le \|X - XEFZ\|^2 + \|NFZ\|^2$$

Summarizing the above,

$$\left\|X - XEF\hat{G}\right\|^2 + \left\|NF\hat{G}\right\|^2 \leq \|X - XEF\hat{G}\|^2 + \|NEF\hat{G}\|^2 \leq \|X - XEF\hat{G}\|^2 + \|NF\hat{G}\|^2$$

$$\left\|X - XEF\hat{G}\right\|^2 + \left\|NF\hat{G}\right\|^2 = \|X - XEF\hat{G}\|^2 + \|NEF\hat{G}\|^2$$

$$\left\|NF\hat{G}\right\|^2 = \|NEF\hat{G}\|^2$$

$N$ being full rank, this yields $EF\hat{G} = F\hat{G}$.

Replacing into (14), and setting $H = EFG$, we have

$$\hat{G} = \arg\min_G \|X - XEFG\|^2 + \|NFG\|^2$$

$$= \arg\min_G \|X - XEFG\|^2 + \|NEFG\|^2$$

$$\hat{H} = \arg\min_H \|X - XH\|^2 + \|NH\|^2$$

Finally, $E\hat{H} = EEF\hat{G} = EF\hat{G} = \hat{H}$, since $E$, a binary diagonal matrix, is involutive. This completes the proof. $\qquad\square$

## A.2 MODELING MEASUREMENT NOISE

Equation 1 does not explicitly contain a measurement noise term. Yet, in moast experimental cases, the problem is best described as:

$$Y = (XE + N)F + M \tag{16}$$

with $M \in \mathbb{R}^{n \times d_y}$.

This equation is actually equivalent to Equation 1 given our hypotheses. Indeed, we can rewrite $M = MF^{-1}F$ over $Img(F)$, which leads to:

$$Y = (XE + N)F + M = (XE + N + MF^{-1})F = (XE + N')F$$

Consequently, assuming that $F$ is full rank on $Img(XE)$, B2B yields the same solutions to equations 1 and 16.

## A.3 FEATURE IMPORTANCE

For B2B, feature importance is assessed as follows:

---
**Algorithm 2:** B2B feature importance.

**Input:** $X_{train} \in \mathbb{R}^{n \times d_x}$, $X_{test} \in \mathbb{R}^{n' \times d_x}$, $Y_{train} \in \mathbb{R}^{n \times d_y}$, $Y_{test} \in \mathbb{R}^{n' \times d_y}$,
**Output:** estimate of prediction improvement $\Delta R \in \mathbb{D}^{d_x}$.

1  $H, G = $ B2B$(X_{train}, Y_{train})$;
2  $R_{full} = $ corr$(X_{test}H, Y_{test}G)$;
3  **for** $i = 1, \ldots, d_x$ **do**
4  $\quad$ $K = Id$;
5  $\quad$ $K[i] \leftarrow 0$;
6  $\quad$ $R_k = $ corr$(X_{test}KH, Y_{test}G_i)$;
7  $\quad$ $\Delta R_i = R_{full} - R_k$;
8  **end**
9  **return** $\Delta R$

---

For the Forward Model, the feature importance is assessed as follows:

---
**Algorithm 3:** Forward feature importance.

**Input:** $X_{train} \in \mathbb{R}^{n \times d_x}$, $X_{test} \in \mathbb{R}^{n' \times d_x}$, $Y_{train} \in \mathbb{R}^{n \times d_y}$, $Y_{test} \in \mathbb{R}^{n' \times d_y}$,
**Output:** estimate of prediction improvement $\Delta R \in \mathbb{D}^{d_x, d_y}$.

1  $H = $ LinearRegression$(X_{train}, Y_{train})$ $R_{full} = $ corr$(X_{test}K, Y_{test})$;
2  **for** $i = 1, \ldots, d_x$ **do**
3  $\quad$ $K = Id$;
4  $\quad$ $K[i] \leftarrow 0$;
5  $\quad$ $R_k = $ corr$(X_{test}KH, Y_{test})$;
6  $\quad$ $\Delta R_i = R_{full} - R_k$;
7  **end**
8  **return** $\Delta R$

---

For the CCA and PLS models, the feature importance is assessed as follows:

---

**Algorithm 4:** CCA and PLS feature importance.

**Input:** $X_{train} \in \mathbb{R}^{n \times d_x}$, $X_{test} \in \mathbb{R}^{n' \times d_x}$, $Y_{train} \in \mathbb{R}^{n \times d_y}$, $Y_{test} \in \mathbb{R}^{n' \times d_y}$,
**Output:** estimate of prediction improvement $\Delta R \in \mathbb{D}^{d_x, d_z}$.

1   $H, G = \text{CCA}(X_{train}, Y_{train})$;
2   $R_{full} = \text{corr}(X_{test} H, Y_{test} G)$;
3   **for** $i = 1, \ldots, d_x$ **do**
4      $K = Id$;
5      $K[i] \leftarrow 0$;
6      $R_k = \text{corr}(X_{test} K H, Y_{test} G)$;
7      $\Delta R_i = R_{full} - R_k$;
8   **end**
9   **return** $\Delta R$

---

For the Backward Model, feature importance cannot be assessed because there is no prediction.

### A.4   RECOVERING E

In case of noise, B2B yields non binary $\hat{E}$. Three thresholding rules can be used to binarize its values thus explicitly recover causal features.

First, given known signal-to-noise ratio, the threshold above which a feature should considered to be causal can be derived analytically. Indeed, Equation 9 implies that the $k$ first diagonal elements of $\hat{H}$ are bounded:

$$0 \leq \frac{\sigma_{X_k}}{\sigma_{X_k} + \sigma_{N_1}} \leq \text{diag}_k(\hat{H}) \leq \frac{\sigma_{X_1}}{\sigma_{X_1} + \sigma_{N_k}}$$

where $\sigma_{X_1}$, $\sigma_{X_k}$, $\sigma_{N_1}$ and $\sigma_{N_k}$ denote the largest and smallest eigenvalues of $\Sigma_{X_1 X_1}$ and $\Sigma_{N_1 N_1}$.

The average value $\mu$ of non-zero coefficients of $\text{diag}(\hat{H})$ is the trace of $\hat{H}$ divided by $k$, and can be computed as

$$\mu = \frac{Var(X)}{Var(X) + Var(N)} \tag{17}$$

The decision threshold between causal and non-causal elements is thus a fraction $\mu$, whose proportion arbitrarily depends on the necessity to favor type I and type II errors. In practice, we cannot use this procedure for our MEG study, because signal-to-noise ratio is unknown.

Second, $\text{diag}(\hat{H})$ can be binarized with the Sonquist-Morgan criterion (Morgan & Sonquist, 1963), a non-parametric clustering procedure separating small and large values in a given set. This procedure maximizes the ratio of inter-group variance while minimizing the intra-group variance, over all possible splits of the diagonal into $p$ largest values and $d_x - p$ smallest values. Let $m_0$ and $m_1$ be the average values of the two clusters, $p$ and $d_x - p$ their size, and $v$ the total variance of the sample, Sonquist-Morgan criterion maximizes (Kass, 1975):

$$\frac{p(d_x - p)}{d_x} \frac{(m_1 - m_0)^2}{v} \tag{18}$$

This procedure assumes that there exists at least one causal and at least one non-causal feature. Third, second-order statistics across multiple datasets can be used to identify the elements of $\text{diag}(\hat{H})$ that are sgnificantly different from 0. This procedure is detailed in the method section of our MEG experiment.

Overall, these three procedure thus varies in their additional assumptions: i.e. (1) a known signal-to-noise ratio, (2) the existence of both causal and non-causal factors or (3) independent repetitions of the experiment.

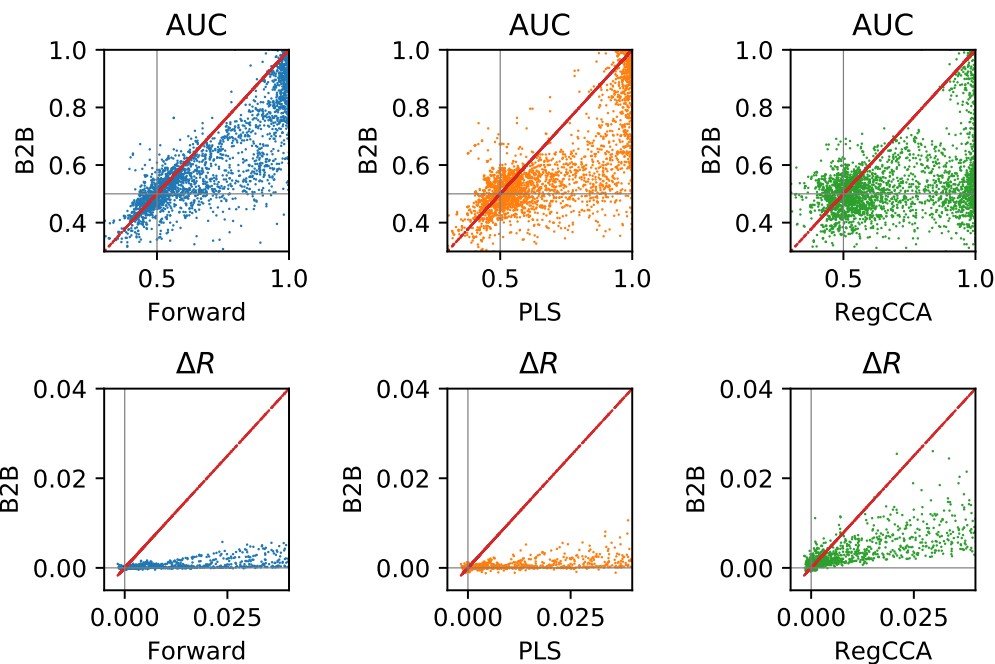

Figure 5: Synthetic experiments. Distribution (over conditions) of AUC (top) and Feature Importance $\Delta R$ (bottom) metrics between our method (y-axis) and the baselines (x-axis). Each dot is a distinct synthetic experiment. Dots below the diagonal indicates that B2B outperform the tested model.

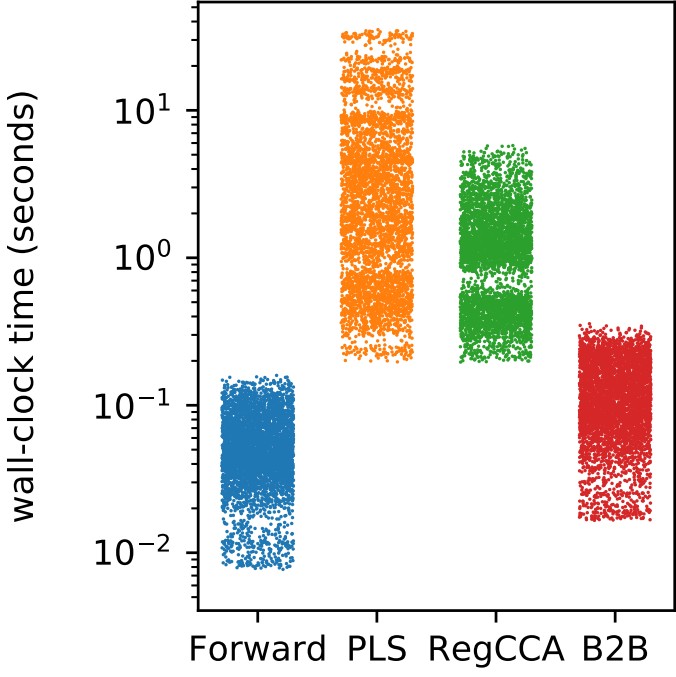

Figure 6: Wall-clock run-time for our method B2B and for the baselines. Each dot is a distinct synthetic experiment. B2B runs much faster than cross-decomposition baselines.

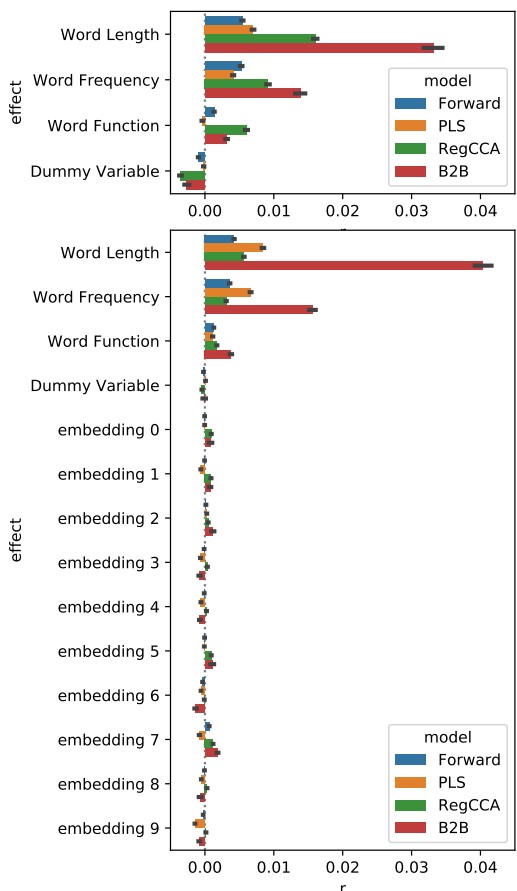

Figure 7: Comparison of $\Delta R$ when the models are tested on four variables (top) and when the models are tested on an these four variables as well as another 10 word-embedding features (bottom). These results illustrate that, unlike Regularized CCA, B2B remains robust even when the number of tested factors increases.

## B  ADDITIONAL FIGURES

### B.1  ROBUSTNESS TO INCREASING NUMBER OF FACTORS

To test whether each of the methods robustly scales to an increasingly large number of potential causes $X$, we enhanced the four ad-hoc features (word length, word frequency, word function, dummy variable) with another ten features. These additional features corresponds to the first dimensions of word embedding as provided by Spacy (Honnibal & Montani, 2017). The results shown in Figure 7, show that the feature importance of ad-hoc features as derived by B2B remain unchanged and are actually improved.

