# OpenReview forum: "Measuring causal influence with back-to-back regression: the linear case"
_ICLR.cc/2020/Conference — Reject_

### Official Review · AnonReviewer1 · 2019-10-17
**Official Blind Review #1**

**Rating:** 3

**Review:**


The paper provides an iterative linear method to identify causal influences of putative cause matrix to signal matrix. The idea is a natural extension of previous forward and backward such as CCA and PLS. The paper has provided consistency guarantee and several synthetic and real data experiments as support.

Technical questions:

1) The estimation of binary causal influence matrix E is set as \hat{E} = Diag(\hat{H}). Why is Diag(\hat{H}) guaranteed to have binary elements?

2) The Theorem 1 needs more explanation about why it proves consistency, which is currently isolated from other parts of the paper. Why E\hat{H} = \hat{H} guarantees the consistency of \hat{E} = Diag(\hat{H})?  For example, \hat{H} can have more all-zero rows than E which still satisfies E\hat{H} = \hat{H}  but  \hat{E}  is not equals to E. In an extreme case,  \hat{H} = 0 will have  E\hat{H} = \hat{H} but is clearly not consistent.

3) How does the Eq (4),(5) give an estimation \hat{E}?

4) By Eq. (10,11), H and G seem to be determined give X,Y. Then what are the maximization of Eq. (12,13) over?


General comments:

1) How does the problem in Eq.(1) differ from variable selection in linear regression where a plenty algorithms exist such as LASSO, spike-slab prior, SCAD, etc. ?

2) The experiments are a bit weak with a simple synthetic experiment and a real dataset with just four features. Can the experiment directly demonstrate the correctness of the theorem?

Typo:

Page 1, “are are based on”

In general, I think B2B is an algorithm that has improvement over CCA and PLS. I am looking forward to the author response to address my above concerns.

##############
I have read the author's feedback which addresses some of the confusion parts in the paper. I maintain the current rating mainly because of the experimental strength.

**Experience Assessment:**

I have read many papers in this area.

**Review Assessment: Checking Correctness Of Derivations And Theory:**

I assessed the sensibility of the derivations and theory.

**Review Assessment: Checking Correctness Of Experiments:**

I carefully checked the experiments.

**Review Assessment: Thoroughness In Paper Reading:**

I read the paper thoroughly.

---

> ### Author Response · Authors · 2019-11-14
> **Reply to R1: general comments**
>
> # (1) How does the problem in Eq.(1) differ from variable selection in linear regression where a plenty algorithms exist such as LASSO, spike-slab prior, SCAD, etc. ?
>
> Variable selection methods are essentially unidimensional in Y. For example, linear regression selects a subset of causes of X for each dimension of Y independently. This is what the Forward models does : the model would be equivalent if each dimension of Y were considered independently.
>
> In contrast, B2B and other cross-decomposition models such as CCA and PLS work with multidimensional X and Y.
>
> B2B, unlike CCA and PLS, efficiently provides interpretable coefficients for causal discovery.
>
>
> # (2) The experiments are a bit weak with a simple synthetic experiment and a real dataset with just four features. Can experiment directly demonstrate the correctness of the theorem?
>
> We generally understand your concern, and in the following we hope to convince you that our experimental setup is sound. On a high level, our reasoning about this method and is the following:
>
> We state and prove a consistency theorem in the linear case (where using theory is possible with our current mathematical knowledge). We believe that the proof is correct (we hope you can help us find mistakes if any), and we believe that the theorem is correct as a consequence. This theoretical study allows us to understand the behavior of this method better.
>
> We then compare our method against baselines on controlled synthetic experiments varying parameters of the problem over a large grid of 25,000 combinations, in order to check that our approach holds, where it succeeds and where it can fail. We believe that our study on synthetic data is thorough. The simplicity of Figure 2, showing merely an aggregate of our results, contrasts with the actual complexity of the study.
>
> We finally apply our method on a real case with non-linear data, using actual brain recordings of human subjects (brain response is non-linear). There, we focus only on a few features, not because our method does not allow using more, but because those features were thoroughly studied in neuroscience: our goal is to compare our conclusions with verified reference points in existing literature, and obtain results assessing the validity of our method and baselines.
>
> Moreover (and we may be to blame for not emphasizing it enough in the manuscript), our experiment is one of the largest ever run : most neuroimaging studies investigate approximately 20 subjects (we have ~100), and use a single analytical method (we study several), generally applied to a single factor of interest.

---

> ### Author Response · Authors · 2019-11-14
> **Reply to R1: technical comments**
>
> # (1) Why is $Diag(\hat{H})$ guaranteed to have binary elements?
>
> In the linear case, $diag(H)$ is binary only in the absence of noise $N$ (more precisely, of noise on the active causal features).
>
> Without loss of generality, we suppose that the non-zero features of $E$ are the $k$ first.
>
> Since $EH = H$ (Theorem), the last $n-k$ rows of $H$ are zero, and so are the last $n-k$ elements of $diag(H)$.
>
> We denote $X_1$ and $N_1$ as the first k features of $X$ and $N$. Therefore, the top left submatrix of $H$ is $Cov(X_1,X_1) (Cov(X_1, X_1)+Cov(N_1,N_1)$ (Eq 8). The diagonal elements of this submatrix will be in $[0,1]$. In the absence of noise on the k first features, this submatrix is identity.
>
> Therefore, $diag(H)$ is binary.
>
> In the presence of noise, the n-k last terms in $diag(H)$ will remain zero, but the average of the k first will be equal to $Var(X_1) / (Var(X_1)+Var(N_1))$
>
> We now added clarifications at the end of Section 2.2 and in the appendix.
>
>
> # (2) The Theorem 1 needs more explanation about why it proves consistency, which is currently isolated from other parts of the paper. Why E\hat{H} = \hat{H} guarantees the consistency of \hat{E} = Diag(\hat{H})?  For example, \hat{H} can have more all-zero rows than E which still satisfies E\hat{H} = \hat{H}  but  \hat{E}  is not equals to E. In an extreme case,  \hat{H} = 0 will have  E\hat{H} = \hat{H} but is clearly not consistent.
>
> The possibility that $\hat{H}$ has more all-zero rows than E violates one of our assumptions, namely that “$X$ and $F$ are full rank on $Img(E)$” (the subspace spanned by the columns of E).
>
> Indeed, full rank of $X$ over $Img(E)$ implies that $Cov(X_1,X_1)$ has full rank. Similarly $Cov(X_1,X_1) (Cov(X_1,X_1) Cov(N_1,N_1))^{-1}$ is full rank too. Consequently, from Eq (9), none of the first k diagonal elements can be equal to zero.
>
> In layman terms, our hypothesis implies that $E$ can only be recovered if all of its active elements lead to a change in some dimension(s) of $Y$. Otherwise, these elements will be estimated to be non-causal, as expected.
>
> # (3) How does the Eq (4),(5) give an estimation \hat{E}?
>
> The estimation of E derives from Eq (2) and (3). In contrast, Eq (4) and (5) implement the optimization technique described by Rifkin and Lippert 2007 in order to efficiently estimate optimal L2-regularization parameters through leave-one-sample-out cross-validation over the training set. We clarify this passage in the updated manuscript, in Section 2.1.
>
> # (4) By Eq. (10,11), H and G seem to be determined give X,Y. Then what are the maximization of Eq. (12,13) over?
>
> We apologize for our confused notation. Equation (10) describes the calculation of a forward model.
> Equation (11) describes the  calculation of a backward model.
> Equation (12) describes CCA.
> Equation (13) describes PLS
> In other words, G and H represent different matrices in each equation. We used similar notations to highlight the functional similarities of these matrices in the different models; but in order to avoid any confusion, we now add indices (e.g. $H_{cca}$, $G_{pls}$) to emphasize their differences (see equations in Section 3.2).

---

### Official Review · AnonReviewer3 · 2019-10-23
**Official Blind Review #3**

**Rating:** 3

**Review:**

The authors introduce a method (B2B) for disentangling effects of correlated predictors in the context of high dimensional outcomes. Their model assumes the outcomes are constructed by a linear transformation of a set of true causes plus measurement noise. Specifically, they fit the model Y=(XE+N)F, where X are the predictors, E is a binary matrix indicating the true causes, N is a noise term, and F is a mixing term. They provide a closed form solution the model fit based on a pair of l2-regularized regressions. They simulate from the given model and provide comparisons against least-squares regression, canonical correlation analysis, and partial least squares. They also apply the method to brain imaging data containing 102 individuals reading 120 sentences plus scrambled sentences, with the goal of inferring which features of the words have an effect on imaging results.

This paper appears to be technically sound, but it should be rejected based on 1) the relatively limited applicability of the model and 2) a lack of thorough experimentation indicating that this is an appropriate method under more general circumstances. It is odd that the model assumes the outcomes are measured without error. Instead, it is assumed that the causes are measured with error, and mixed via F. A more appropriate model may be: Y=(XE+N)F+M, where an additional noise term M allows for Y to be measured imprecisely. Viewed in this light, the model starts to look a lot like canonical correlation analysis. Consider a model Y=ZF+M, X=ZG+N, if dim(Z) = dim(X) and G is invertible, this can be re-written as Y=(X inv(G)-N)A+M, and we arrive at a similar model with specific restrictions about the structure of inv(G) (E will in general not be invertible, so they are not the same). It is particularly odd, then, that the authors do no comparisons against any regularized form of CCA, which would seem to be the most natural method to use in this circumstance. Moreover, in the simulations where they show B2B outperforms CCA, they use 1000 training samples with 10-100 possible causes. In their experiments on real data, where it seems the CCA results are much closer to the results that B2B gives, they have 2700 samples and 4 possible causes. This seems to imply the real data case might be better conditioned than the simulated case, so that regularization would have less of an impact.

In conclusion, the authors present a sound method for disentangling correlated possible causes when the outcome is high-dimensional. However, the authors do not provide enough evidence that this method is generally useful and better than established methods to merit acceptance to ICLR. A comparison to regularized CCA, application to more datasets and simulations under violations of the model would greatly improve the paper. I also have two minor points. 1) the word “causal” can mean many things, and here it refers specifically to disentangling correlated predictors, rather than confounding in observations or direction of effect. It would improve the paper to add some discussion of this point. 2) The comparisons in the experiments are done between E estimated from B2B and E=sum_j H_j^2 for other methods that do not directly estimate E. However, a more natural comparison might be against EF as this also includes estimates of the strength of influence of each observation, which is implicit in the sum above.


**Experience Assessment:**

I have read many papers in this area.

**Review Assessment: Checking Correctness Of Derivations And Theory:**

I assessed the sensibility of the derivations and theory.

**Review Assessment: Checking Correctness Of Experiments:**

I assessed the sensibility of the experiments.

**Review Assessment: Thoroughness In Paper Reading:**

I read the paper thoroughly.

---

> ### Author Response · Authors · 2019-11-14
> **Reply to R3: part 1**
>
> # (1) This paper appears to be technically sound, but it should be rejected based on 1) the relatively limited applicability of the model and 2) a lack of thorough experimentation indicating that this is an appropriate method under more general circumstances.
>
> We agree with R3 that the present paper focuses on a specific issue, originally motivated by a precise empirical problem: i.e. finding, among multiple competing factors, those that impact a noisy system measured with multiple channels.
>
> However, we partially disagree on “the limited applicability” of our method. Disentangling causes under noisy multidimensional observations is pervasive in observational studies (e.g. Detecting gravitational waves necessitate analytical solutions to disentangle confounding heterogeneities) as well as experimental studies, where cause disentanglement is often restricted to factorial designs (e.g. collecting brain responses to words carefully selected such that they are matched in length across different frequencies, an approach that does not scale to increasingly numerous parameters).
>
> Here, we provide, with theoretical guarantees, a solution to this general issue in the linear case.
>
> We demonstrate the usability of our method both in a variety of synthetic experiments and confirm that B2B can (1) systematically outperform baseline methods and (2) be robust to numerous factors. Furthermore, we demonstrate the usability of our method by analyzing an exceptionally large dataset (100 MEG subjects, where most neuroimaging studies are recording 15-20 subjects) and we now provide additional analyses to assess the robustness of B2B when additional word-embedding features are included in the analysis (Appendix: Robustness to increasing number of factors).
>
> While we agree with R3 that that it would be best to show that B2B effectively address disentanglement in an additional experimental setups, we believe that providing our solution to the community is the first step to achieve this objective.
>
>
> # (2) It is odd that the model assumes the outcomes are measured without error. A more appropriate model may be: Y=(XE+N)F+M
>
> This is a good point which we insufficiently detailed in our original submission. We agree with R3 that measurement noise M is likely: i.e.
>
> $Y = (XE + N) F + M$
>
> where F represents the unknown response function of the measurement apparatus, and N corresponds to noise before measurement (e.g. background brain activity, eye movements) and M corresponds to measurement noise (e.g. bad sensor, electronic irregularities etc).
>
> However, we can rewrite $M$ as $M’F$ over $Img(F)$, the subspace spanned by the columns of $F$. Note that by hypothesis, $Img(E)$ is included in $Img(F)$. Furthermore, by definition, $F$ is full rank over $Img(F)$. Therefore, $M’ = M F^{-1}$, which yields:
>
> $Y = (XE + N) F + M = (XE + N) F + M’ F = (XE + N + M’) F = (XE + N’) F$
>
> Consequently, measurement noise $M$ is absorbed by $N F$ when these two matrices are unknown.
>
> We added this clarification to the manuscript (see Appendix: Modeling measurement noise).
>
> # (3) The model starts to look a lot like canonical correlation analysis.
>
> We agree with R1, that B2B and CCA share a common ground. Specifically, in the basic non-regularized case, both B2B and CCA compute :
>
> $O = (X’ X)^{-1} XY (Y’Y)^{-1} YX$
>
> CCA additionally perform an eigen decomposition of O in order to identify the orthogonal components where X and Y are maximally correlated (a.k.a canonical components).
>
> B2B additionally extracts the diagonal of O, to recover E (the non-invertible component of the X to Y mapping).
>
> Therefore, B2B and CCA have different objectives.
>
> CCA could be adapted to perform causal estimation. However, this must be performed in canonical space, and thus across all canonical components. In contrast, B2B performs causal estimation in feature space. As a result (i) B2B is more interpretable and (ii) does not dilute causal estimation over several dimensions.
>
> Finally, B2B allows several technical and computational improvements such as (i) the necessity to use bagging between the two regressions, (ii) the possibility to vary regularization parameters for each of the two regressions, and (iii) the use of using computationally efficient grid search (Eq. 4-5).
>
> We now updated the discussion to clarify the similarities and differences between B2B and CCA.

---

> ### Author Response · Authors · 2019-11-14
> **Reply to R3: part 2**
>
>
> # (4) The authors do no comparisons against any regularized form of CCA
>
> We thank R3 for this comment. We now revised the manuscript (see Section 3, both synthetic and MEG experiments) to change CCA into an l2-regularized CCA, as implemented by the Pyrcca package provided by Bilenko and Gallant (2016). L2-regularization CCA is now optimized similarly to B2B, i.e. over a nested-grid search optimization of the training set over 20 values logarithmically distributed between 1e-4 and 1e4. Overall, our updated results do not change the conclusion of our paper. However, we do observe one experimental case where regularized CCA outperforms B2B: the feature importance of the word function effect in MEG is higher with regularized CCA than with B2B. This unexpected superiority of CCA over B2B disappears when more than 4 features are tested.
>
> # (5) The real data case might be better conditioned than the simulated case
>
> We agree with R1 that the optimal use-case for B2B appears to be when a large number of covarying factors are investigated, as demonstrated in the synthetic experiments.
>
> In this first method paper, we aimed to verify that B2B yields to plausible results. We thus intentionally investigated well-described phenomena (the neural correlates of word length and word frequency). B2B successfully matched our expectations. First, word length and word frequency revealed early and late brain effects respectively. Second, B2B did not reveal any spurious effect before stimulus onset. Third B2B appeared reliably better than other baseline methods.
>
> To address R1 comments, we now added an additional MEG analysis (see Figure 7 in the appendix) in which introduce additional features: the word-embedding vectors of each word provided by the Spacy package. Our results show that B2B remains robust to the introduction of additional factors.
>
>
> # (6) The word “causal” can mean many things, and here it refers specifically to disentangling correlated predictors, rather than confounding in observations or direction of effect. It would improve the paper to add some discussion of this point.
>
> We thank R3 for this remark. We now clarify the definition in the manuscript:
>
> “
> The present paper focuses on the restricted issue of disentangling the causal influence of linearly correlated predictors ($X$) onto multivariate observations ($Y$). The present approach thus differs from other causal discovery algorithms based on temporal-delays and/or nonlinear interaction in systems where the directionality of causation (from X to Y or vice versa) is unknown (e.g. \citep{peters2017elements, granger1969investigating, janzing2013quantifying, scholkopf2016modeling}.
> “
>
>
> # (7) The comparisons in the experiments are done between E estimated from B2B and E=sum_j H_j^2 for other methods that do not directly estimate E. However, a more natural comparison might be against EF as this also includes estimates of the strength of influence of each observation, which is implicit in the sum above.
>
> We agree that EF is likely to be closer to Sum_j H_j^2. However, the precise purpose of B2B, unlike other methods, is to retrieve E when F is unknown. The introduction of Sum_j H_j^2 is solely designed to provide a fair chance to previous baseline.

---

### Official Review · AnonReviewer2 · 2019-10-27
**Official Blind Review #2**

**Rating:** 6

**Review:**

This paper proposes "Back-to-Back" regression for estimating the causal influence between X and Y in the linear model Y=(XE+N)F, where the E denotes a diagonal matrix of causal influences. Furthermore, this work theoretically shows the consistency of B2B and the experiments also verify the effectiveness of this approach.

The writing is well and clear and there are some minors issues:
- Further analysis and explanation for using \hat{E}=Diag(H)to estimate the causal influence might be needed.
- The model defined in Fig. 1 seems the influence E should have a sparse diagonal vector. It is possible to introduce an L1 regulation in E?

##############
After reading the author's feedback and the comments from other reviewers, I keep the current rating but tend to a borderline score and it is ok if it must be rejected because of the concerns of limited applicability and the experimental.


**Experience Assessment:**

I have read many papers in this area.

**Review Assessment: Checking Correctness Of Derivations And Theory:**

I assessed the sensibility of the derivations and theory.

**Review Assessment: Checking Correctness Of Experiments:**

I assessed the sensibility of the experiments.

**Review Assessment: Thoroughness In Paper Reading:**

I read the paper at least twice and used my best judgement in assessing the paper.

---

> ### Author Response · Authors · 2019-11-14
> **Reply to R2**
>
> # (1) clarify $\hat{E}=Diag(H)$
>
> We agree with #R2 that the original manuscript insufficiently detailed the relationship between diag(H) and the causal estimates.
>
> We have now expanded the proof and added a section in the Appendix to clarify how E can be recovered from $diag(\hat H)$.
>
> In addition, we describe in the appendix three methods to binarize diag(H) into causal and non causal features, when (1) signal to noise ratio is known (2) independent experiments are repeated or (3) we know that X contains both causal and non-causal features.
>
>
> # (2) Can E use a sparse prior?
>
> E should tend towards a sparse diagonal vector when a small proportion of factors causally influence Y.
>
> In our implementation, B2B uses L2-regularization in both the forward regression (H) and the backward (G) regressions. However, any regularization can be used.
>
> Note that a distinct regularization can be implemented and optimized for each regression separately (e.g. L2 for G and L1 for H). In this regard, we did pilot with L1-regularization on the H regression, to induce sparsity in E. However, we did not observe any clear improvement on our synthetic or MEG experiments, and this approach was significantly less efficient computationally. Indeed, the efficient leave-one-out optimization of l2-regularization parameters detailed by Rifkin and Lippert 2007 only applies to l2 regularization.
>
> Finally, sparsity can be a posteriori enforced onto $\hat E$ via a thresholding method. As mentioned above, we describe three thresholding methods in the appendix, together with their respective assumptions.

---

### Author Response · Authors · 2019-11-14
**Response summary**

We thank our three reviewers for their careful reviews and helpful comments.  The main changes of our manuscript are:
A clarification and substantial expansion of the proof of our theorem
A successful comparison of B2B against l2-regularized CCA for both synthetic and MEG experiments, and a clarification of its link with B2B
A simplification of the feature importance algorithm
Additional tests of B2B and other baseline models when the MEG recordings are modeled with a larger set of potential factors derived from word-embeddings.

Overall, these additional analyses and clarification confirm that B2B is an efficient method to disentangle the causal influence of linearly correlated predictors ($X$) onto noisy multivariate observations ($Y$).

---

### Decision · Program_Chairs · 2019-12-19

**Decision:**

Reject

**Comment:**

The authors introduce a method for disentangling effects of correlated predictors in the context of high dimensional outcomes. While the paper contains interesting ideas and has been substantially improved from its original form, the paper still does not meet the quality bar of ICLR due to its limitations in terms of limited applicability and experiments. The paper will benefit from a revision and resubmission to another venue.